# Comparison of the Effects of Constraint-Induced Movement Therapy and Unconstraint Exercise on Oxidative Stress and Limb Function—A Study on Human Patients and Rats with Cerebral Infarction

**DOI:** 10.3390/brainsci13010004

**Published:** 2022-12-20

**Authors:** Dong Wang, Lijuan Li, Hongxia Pan, Liyi Huang, Xin Sun, Chengqi He, Quan Wei

**Affiliations:** 1Rehabilitation Medicine Center and Institute of Rehabilitation Medicine, West China Hospital, Sichuan University, Chengdu 611135, China; 2Key Laboratory of Rehabilitation Medicine in Sichuan Province, Sichuan University, Chengdu 611135, China; 3Department of Rehabilitation Medicine, Affiliated Hospital of Chengdu University, Chengdu 610081, China

**Keywords:** CIMT, exercise, function, oxidative stress, cerebral infarction

## Abstract

Most conventional post-stroke rehabilitation treatments do not involve imposed constraints of the unaffected limb. In contrast, Constraint-Induced Movement Therapy (CIMT) is comprised of massed task practice with the affected limb and constraint of the unaffected limb. CIMT is a promising rehabilitation technique used for motor recovery of affected limbs after stroke, but its effectiveness and mechanism are not fully understood. We compared the effects of the two exercise modes on limb function post-stroke in animal models and human subjects, and investigated whether oxidative stress response was involved in regulating the effects. We first conducted a randomized controlled trial (RCT), in which 84 subjects with cerebral infarction were assigned to dose-matched constraint-induced movement therapy (CIMT), or unconstraint exercise (UE), or conventional rehabilitation treatment. Motor functions of the limb are primary outcomes of the RCT measured using Brief Fugl–Meyer upper extremity score (FMA-UE), Ashworth score, and Barthel scale. Psychological influence of CIMT and UE was also examined using Self-Rating Depression Scale (SDS). Next, we investigated the effects of CIMT and UE in rats undergoing middle cerebral artery occlusion and reperfusion (MCAO/R). Motor function, infarct volume, and pathohistological changes were investigated by mNSS, MRI, and histological studies. The role of Keap1-Nrf2-ARE was investigated using qRT-PCR, Western blot, immunochemistry, immunofluorescence, and ELISA experiments. In RCT, patients taking CIMT had a higher score in FMA-UE, Barthel index, and SDS, and a lower score in modified Ashworth, compared to those taking UE. In rats receiving CIMT, motor function was increased, and infarct volume was decreased compared to those receiving UE. The expression of Keap1 protein and mRNA in the peri-infarct tissue was decreased, and Nrf2 and ARE protein and mRNA were increased in rats receiving CIMT compared with UE. Nrf2 agonist t-BHQ increased the benefits of CIMT. In conclusion, CIMT is more effective than UE in improving upper limb motor function, reducing muscle spasm in patients with cerebral infarction compared to UE, but patients receiving CIMT may feel depressed. Moreover, both CIMT and UE are beneficial to limb function recovery and limit the infarct expansion in MCAO/R rats, but CIMT was more effective than UE. Oxidative stress reaction has an essential role in regulating the CIMT induced benefits.

## 1. Introduction

Stroke is the second leading cause of death and disability worldwide [1]. The high morbidity, mortality, and disability rates of stroke pose a strenuous burden to society, families, and patients. The number of new stroke patients is about 13 million worldwide every year, and its incident rate is still on the rise. At present, the incidence of stroke in China is increasing at an annual rate of 8.7% [2,3]. According to previous research, up to 80% of stroke survivors develop hemiparesis, and 40% suffer from upper limb dysfunction hampering daily activity and quality of life [4,5]. Structured rehabilitation is recommended for stroke patients and an essential component of rehabilitation is exercise therapy. Exercise can improve motor function and daily activity skills of stroke survivors, increase the quality of life, and reduce the risk of stroke recurrence [6,7]. However, there are many modes of exercise therapies currently used in clinical practice with different indications [8,9,10]. There is a lack of consensus on long-term clinical efficacy and mechanism of action for these exercise therapies, which poses difficulties to the implementation of exercise therapy in clinical practice.

Constraint-induced movement therapy (CIMT) has been developed as a rehabilitation exercise to improve the function of the upper limb in survivors of stroke [11,12]. The therapy is a behavioral approach based on the principle of “learned non-use”, which was first described by Taub [13]. Learned non-use is a clinical term that refers to a motor deficit after nervous system damage caused by learned suppression use of the limb, reinforced over time by poor quality movement [14]. Generally, CIMT discourages learned non-use by constraint devices worn by the patients on the unimpaired side, and is often accompanied by intensive repetitive task practice using the unconstrained impaired limb.

Treadmill training is an unconstraint exercise (UE) mode commonly practiced to improve limb function, which has a higher exercise intensity and lower frequency compared to CIMT [15,16]. The comparison of effects of CIMT and UE in improving limb function has not been carried out by previous studies.

Oxidative stress reaction is the imbalance of synthesis and clearance of reactive oxygen species (ROS), causing peroxidation of biomacromolecules such as nucleic acids, lipids, and proteins. It can lead to the production of toxic substances, thereby causing cell dysfunction, damage, or apoptosis [17]. ROS produced by the cell damage, such as hydrogen peroxide (H_2_O_2_), superoxide anion (O^2−^), and hydroxyl radicals (-OH), further destroys cell functions. Increasing studies have confirmed that oxidative stress is one of the important pathological mechanisms of neuronal cell damage and death after stroke [18]. Ischemia and hypoxia after stroke can induce oxidative stress damage, calcium overload, inflammatory cascade, and excitatory amino acid toxicity, causing nerve cell death and serious damage to the brain [19,20]. There exists a link between exercise-induced recovery of neural function and oxidative stress reaction [21,22]. Given the relationship between exercise and ROS and the positive effect of exercise training on neural recovery, it is intriguing how different exercise modes influence oxidative stress reaction after stroke.

Here, we conducted an RCT and animal study to evaluate the effects of two exercise modes: CIMT and UE, in limb dysfunction of stroke survivors, and to investigate the relationship between CIMT/UE and oxidative stress reaction. Our study attempts to provide evidence for the clinical practice of stroke rehabilitation.

## 2. Methods and Materials

### 2.1. Study Population of the RCT

The protocol of the RCT was approved by the ethical committee of our institution (Year 2020, No. 1011), and the study was registered on Chinese clinical registry (ChiCTR2100042339).

Patients with cerebral infarction were recruited between December 2020 and June 2021, from the Department of Rehabilitation Medicine of West China Hospital, Sichuan University, and the Department of Rehabilitation Medicine of Affiliated Hospital of Chengdu University. Patients who were eligible should age between 30 and 80, and were diagnosed with cerebral infarction in accordance with the Chinese guideline for the diagnosis and treatment of ischemic stroke [23], with a disease course from 10 days to 6 months. In addition, included patients should have unilateral but not bilateral limb dysfunction, with a score of Fugl–Meyer ≥56 (85% of total score) indicating mild to moderate motor impairment. Furthermore, all potential participants were screened regarding whether the included participants can perform the tasks involved in CIMT using the following criteria: extension of wrist joint >20°; extension of metacarpophalangeal joint and interphalangeal joint of the thumb, and at least two of the rest four fingers >10°, and the movement can be repeated three times within 1 min; shoulder joint passive flexion and abduction >90°, external rotation >45°; elbow extension ≤30°; forearm pronation and supination >45°; and the static standing posture can be maintained for at least 2 min [24]. Other inclusion criteria were as follows: have normal consciousness; have no severe cognitive impairment; have stable vital signs; and understand, cooperate, and comply with the planned study procedures.

Patients were excluded if they (1) were in unstable condition, or in the progressive state of stroke, or have cerebral hemorrhage, or (2) had a previous history of stroke, or limb motor dysfunction prior to the onset of cerebral infarction; or (3) had any serious concomitant systemic disease, condition, or disorder that, in the opinion of the investigator, was not appropriate for this study; or (4) had bilateral hemispheric injury.

### 2.2. Randomization and Allocation Concealment in the RCT

If participants were eligible after being screened based on the inclusion and exclusion criteria, they were asked to sign the informed consent and were randomized into the CIMT group, NE group, or control group. Randomized assignments were generated using a random number table and placed in sealed envelopes, which were then opened by an independent study coordinator in consecutive order.

### 2.3. Interventions in the RCT

CIMT group received constraint induced movement therapy plus conventional rehabilitation training; UE group received upper limb task-oriented training plus conventional rehabilitation training; and Control group received conventional rehabilitation training only.

All three groups went through conventional rehabilitation training program. The program comprised joint range of motion training, bed activities and transfer training, locomotor training, and physical factor therapy (e.g., transcranial magnetic stimulation, biofeedback therapy, and low-frequency electrical stimulation therapy) at physical therapists’ discretion. The participants underwent conventional rehabilitation therapy for 30 min, once a day, five times a week, for 3 weeks.

Participants in the UE group received upper limp task-oriented training [25]. The training mainly comprised object grasping activity training, where the patients were required to use their affected arm to grasp or move an object to specific places on a table. The therapists were allowed to adjust the intensity and difficulty of the training according to the patients’ performance. The training lasts for 30 min a time, once a day, five times a week, and for 3 consecutive weeks [26].

Participants in the CIMT group underwent constraint-induced movement therapy. The patients wore mittens for 1 h each time to limit the use of their unaffected hand; thus, they were forced to use affected hands to perform specific tasks. Three to five tasks (e.g., eating, ball throwing, dominoes, playing chess or cards, writing, and sweeping the floor) were selected for each patient based on the functional assessment by the therapist. Before the training, the patients were given instructions on how to use the mitten to improve safety and adherence. The training lasts for 1 h each time, once a day, five times a week, for 3 consecutive weeks [27,28].

### 2.4. Models in the Animal Study

The middle cerebral artery occlusion and reperfusion (MCAO/R) model was used in the animal study. After anesthesia, a middle incision was made in the neck of the rats, and the right common carotid and external carotid arteries were carefully exposed. A suture with a rounded tip coated with poly-L-lysine was inserted approximately 20 mm through the internal carotid arteries. The middle cerebral artery was distally occluded. After 90 min, the suture was withdrawn. Sham operation was conducted by exposing the carotid arteries without occlusion of the middle cerebral artery.

According to our design, we used Nrf2 antagonist ML385, and Nrf2 agonist t-BHQ to evaluate the regulatory role of Nrf2 in mediating CIMT or NE-induced neurological functional recovery. ML385 was dissolved in dimethyl sulfoxide (DMSO) and diluted with saline to 5% DMSO concentration for use. ML385 was injected intraperitoneally with a dose of 30 mg/kg every day, from days 1 to 14 after MCAO/R.

Another group of rats were injected intraperitoneally with t-BHQ. The t-BHQ was dissolved in DMSO and diluted with PBS to 1% DMSO concentration. The single dose was 16.7 mg/kg, and the injections were given every 8 h for 3 times after MCAO.

### 2.5. Interventions in the Animal Study

After successful model establishment, rats were randomly assigned to model group, or CIMT group, or UE group, or normal control (NC) group, or sham group. The interventions began 24 h after MCAO. Rats in the CIMT group were forced to use the affected limbs during the training, by fixation of the unaffected limbs in natural flexion position with a plaster cast. A cotton lining was used to prevent damage to the skin. The animals wore the plaster cast throughout the 14-day intervention phase. To increase the intensity of the CIMT, the rats underwent a 30-min treadmill training every day, and the treadmill speed was set as 8 m/min for days 1–3, 12 m/min for days 4–7, and 15 m/min for days 7–14 [29]. For rats in the UE group, the same treadmill training protocol was used except that their unaffected limbs were not fixed.

### 2.6. Modified Neurological Severity Scores (mNSS)

The mNSS test was performed to evaluate neurological functional outcome on days 1, 4, and 14. The mNSS included motor, sensory, reflex, and balance tests. The score of the mNSS test ranges from 0 to 18, and a higher score indicates a worse neurological function.

### 2.7. Magnetic Resonance Imaging (MRI) of Rats

MRI was used to assess the infarct volume of the brain. Fifteen slices of T2-weighted images were acquired with 0.8 mm thickness, repetition time 2500 msec, and echo time 33 msec. The infarct volume was quantified with the following formula: infarct volume = L_max_ × W × T/2, where L_max_ and W is the maximum and minimum diameter, respectively, of the slice with the largest infarction area, and T is the product of the number of slices and slice thickness.

### 2.8. Histology

The rats were sacrificed and brain tissues were collected for further histological analysis. The brains were perfused with saline and 4% paraformaldehyde, and then were fixed in 10% neutral buffered formalin for 24 h. Then, the brain tissues were dehydrated, embedded in paraffin, and cut into 4 μm slices.

HE staining was performed according to standard protocols. Briefly, after dewaxing with xylene, the slides were stained with hematoxylin for 2 min and thereafter with eosin. After dehydration, neutral gum was used to seal the slides for microscopic analysis.

Nissl staining was performed to assess neurological survival. The slides were deparaffinized and stained with cresyl violet for 1 min. Then, the slides were rinsed with distilled water, dehydrated with alcohol, and mounted with neutral gum for later analysis.

For immunofluorescence (IF), the slides were deparaffinized and rehydrated following standard protocol. Then, the slides were blocked with 10% goat serum at room temperature, and incubated with a primary antibody at 4° overnight, then with a fluorescent secondary antibody for 1 h at room temperature. The nuclei were stained with DAPI. Then, the slides were mounted with anti-quenching medium and were observed under a fluorescent microscope.

For immunohistochemistry (IHC), after dewaxing and rehydration, the slides were immersed in citric acid and heated in micro-oven for antigen retrieval. Endogenous peroxidase activity was blocked by incubation with 0.3% hydrogen peroxide for 30 min. The primary antibody was incubated overnight and the secondary antibody for 1 h. The DAB staining was then performed using a commercial DAB kit. Nuclei were colored with hematoxylin. The slides were sealed and dry for 12 h for microscopic analysis.

### 2.9. QRT-PCR

The border zone tissue was immersed in liquid nitrogen and grounded to powder. Total RNA was extracted using the TRIzol method. The cDNAs were obtained with a reverse transcription kit (Takara) following the manufacturer’s instructions. QRT-PCR was performed with TB GreenTM Premix Ex TaqTM II (Takara) on the CFX96 Real-time PCR Detection System (Bio-Rad). The relative expression of the mRNAs was analyzed by the ∆∆Cq method. The primer sequences are presented in Table 1.

### 2.10. Western Blot

We extracted total proteins from the border zones in the right hemisphere of the brains, and then loaded the protein samples onto an SDS-PAGE gel. After separation by electrophoresis, the protein samples were transferred to polyvinylidene fluoride (PVDF) membranes. Nonspecific binding was prevented using 5% nonfat milk to block the membranes at room temperature for 1 h, followed by incubation with the primary antibodies and the secondary antibody. The immunoblots were detected using an enhancement detection system, and the proteins were quantified with the ImageJ software.

### 2.11. ELISA

Commercial ELISA kit was used to measure levels of MDA in the rat peri-infarct zone of the brain samples. All steps were performed following manufacturer’s instructions. The levels of MAD were expressed as mmol/L.

### 2.12. Statistical Analysis

For the RCT, we calculated the sample size with the Fugl–Meyer score as the primary outcome. Eighty-four participants would be required considering a 1:1:1 allocation ratio, a test power of 80%, a significant level of 5%, a standardized deviation (SD) of 3.4, and a dropout rate of 20%.

Continuous data are presented as the mean ± SD. A rank-sum test or *t*-test were used for intra-group statistical analysis as appropriate; ANOVA or rank-sum test was selected for inter-group comparison as appropriate, with a test level of 0.05. For categorical variables, a Fisher exact probability test was used.

## 3. Results

### 3.1. CIMT Was More Effective Than UE in Improving Limb Function in Stroke Patients

In the RCT, a total of 84 patients were eligible at enrollment. Each group recruited 28 patients, and 3 patients drop out in the control group, 4 in the CIMT group and 2 in the UE group. Nine patients dropped out (5 unscheduled discharged without finishing the treatment; 2 developed severe pneumonia; 2 voluntarily withdrew due to intolerance to constrain). Finally, 75 patients completed the intervention and were analyzed.

The demographic characteristics are depicted in Table 2. At baseline, there was no statistical difference in gender, disease duration, time between disease onset, and rehabilitation treatments among three groups (*p* > 0.05). Figure 1 shows the flowchart of the study.

The baseline FMA-UE, modified Ashworth scale, Barthel index, and SDS were not statistically different among groups (Table 3 and Table 4). Intra-group analysis showed that FMA-UE increased in all three groups (*p* < 0.001), while inter-groups analysis demonstrated that FMA-UE was the highest in the CIMT group compared to the other two groups (*p* < 0.01, Figure 2). Spasm was relieved after receiving CIMT or UE, which was more significant in CIMT, as revealed by the modified Ashworth scale (*p* < 0.01, Figure 2). The Barthel index suggested that both CIMT and UE significantly improved activities of daily living, and CIMT was more effective (*p* < 0.01, Figure 2). CIMT and UE decreased patient depression compared to control intervention, and patients receiving UE had a lower level of depression than those receiving CIMT (*p* < 0.01, Figure 2). The scores of FMA-UE, Ashworth, Barthel, and SDS before and after the interventions are summarized in Table 4 and Table 5.

At baseline, day 7 and day 14, the mNSS test was performed to assess neurological deficits. There was no statistical difference in mNSS scores at baseline and day 7, but at day 14, CIMT had a lower score of mNSS compared to model group or UE group (*p* = 0.004, *p* = 0.026, Figure 3), suggesting that CIMT was more effective in improving motor function. MRI T2WI demonstrated that both CIMT and UE lead to a smaller infarct volume than control (*p* = 0.0001, *p* = 0.032), and CIMT was more effective than UE (*p* = 0.0003, Figure 3).

Hematoxylin–eosin (HE) staining revealed swollen, stained cytoplasm, nuclear pyknosis, nuclear fragmentation, vacuole structure, and disturbed structure in the peri-infarct zone after the induction of MCAO/R, which were alleviated was in the CIMT group compared to control or UE group. Nissl staining suggested an increase in the number of neurons in CIMT group compared with UE or control group (*p* = 0.002, *p* < 0.001, Figure 3).

### 3.2. Keap1-Nrf2 Signaling Pathway Is Linked to the CIMT Induced Benefits

Next, we investigated the role of oxidative stress signaling in the modulation of CIMT or UE induced effects. RT-qPCR suggested that Keap1 mRNA expression was lower in CIMT group than in the UE group (*p* = 0.0004), while the nucleic Nrf2 mRNA level was increased in CIMT compared to the UE group (*p* = 0.0094). Western-blot analysis demonstrated similar changes: that Keap1 protein level was decreased (*p* = 0.0317) and the nucleic Nrf2 protein level was increased (*p* = 0.0174) in the CIMT group. HO-1 was increased in the CIMT group compared with the UE group (*p* = 0.0285). IF and IHC revealed that the protein levels of Keap1 and HO-1 were increased, and Nrf2 was decreased in the CIMT group. The production of MDA was suppressed in the CIMT group, compared with UE or control (*p* = 0.0377, *p* = 0.0194, Figure 4).

### 3.3. Nrf2 Plays a Pivotal Role in Regulating CIMT Induced Benefits

We used Nrf2 antagonist ML385 and agonist t-BHQ to investigate the role of Nrf2. After administration with t-BHQ, the motor function was significantly recovered as suggested by a lower score of mNSS in the CIMT+t-BHQ group than in the model, CIMT, and CIMT + ML385 groups. The CIMT induced motor function recovery was counteracted by ML385. Moreover, the administration of t-BHQ in rats undergoing CIMT resulted in a smaller infarct volume. Rats undergoing CIMT had a smaller infarct volume compared to those receiving CIMT plus ML385.

Using qRT-PCR, Western-blot, and IF, we found that the administration of t-BHQ in rats undergoing CIMT resulted in an increased expression of Nrf2 in protein and mRNA levels. Furthermore, the expression of HO-1 was increased in the CIMT+t-BHQ group compared with CIMT and CIMT+ML385 groups. The injection of t-BHQ in rats undergoing CIMT also resulted in a lower production of MDA (Figure 5).

## 4. Discussion

Our study investigated the clinical efficacy and possible mechanism of constraint and unconstraint exercise mode on limb dysfunction after stroke, in the hope of providing a theoretical basis for the selection of exercise mode in clinical practice. We first conducted an RCT to compare effects of CIMT and UE directly. It was preliminarily found that, compared to UE, CIMT has more significant effects in improving upper limb function and reducing spasm. However, CIMT may cause patients to feel depressed, which may be because some patients did not adapt to the constraint device. Next, in the animal experiment, we studied the improvement of motor function of MCAO/R rats by constraint and unconstraint exercise therapy. Motor function was improved more in CIMT than in UE group as assessed by an mNSS test. Furthermore, the Keap1-Nrf2-ARE signaling pathway was found to play a regulatory role revealed by RT qPCR, Western blot, immunofluorescence, and immunohistochemistry, which was further confirmed in gain of function and loss of function experiments. MDA, the downstream product of oxidative stress, was suppressed in the CIMT and UE group, and CIMT group had a lower level of MDA compared to the UE group. Neural regeneration and apoptosis were also studied, and a higher level of neural regeneration and lower level of apoptosis were observed in the CIMT group than in UE group. Overall, CIMT could significantly improve the motor function of MCAO/R rats, which has a close link to oxidative stress, regeneration, and apoptosis.

Exercise therapy is recognized as an important component of treatment for post-stroke rehabilitation and is crucial in determining patient activity of daily living and quality of life [30]. It is demonstrated that exercise therapy improves limb functional recovery and leads to higher satisfaction, promotes rehabilitation, and lowers health care costs [31,32,33]. Commonly used exercise therapies aiming to improve limb function include CIMT, the motor relearning technique, the Bobath technique, the proprioceptive neuromuscular facilitation (PNF) technique, music supported therapy, the virtual reality technique, and robot-assisted therapy et al., each having its indications or working conditions [34,35]. It is essential to select the appropriate exercise mode for limb function rehabilitation.

Exercise therapy is closely related to oxidative stress. Previous studies have shown that exercise can increase the production of ROS and lead to oxidative stress in many tissues, including blood and skeletal muscle. The main mechanism is that muscle contraction stimulates the production of ROS in active muscle fibers [36,37]. Further studies found that although short time (<1 min) and low intensity (30% maximum oxygen consumption exercise) may not increase oxidative stress, it has been fully confirmed that long-time, high-intensity endurance exercise can lead to the increase of oxidative stress biomarkers in blood and active skeletal muscle in untrained people and animals [38]. On the other hand, research showed that short-term (5 days) and long-term (12 weeks) endurance exercise training increases the activity of antioxidant enzymes in muscles and eliminates the oxidative stress induced by acute exercise. Therefore, the effect of ROS on muscle is bidirectional, depending on exercise intensity, exercise duration, and the level of ROS in muscle fibers [39]. Generally speaking, high-intensity and long-term aerobic exercise produces more ROS than low-intensity, short-term exercise, and can also increase the activity of antioxidant enzymes to eliminate the muscle contraction induced oxidative stress response produced during acute exercise [40]. Therefore, the interaction between exercise and ROS is mutual, and exercise can promote the balance between the rate of ROS production and the rate of ROS scavenging [41].

CIMT is characterized with forced use of impaired limbs to avoid learned non-use phenomenon. Most previous RCTs did not compare CIMT directly with UE, but with conventional treatment protocols instead, and they did not investigate psychological change after wearing the constraint device [11,12,42]. Moreover, in basic research, although previous studies have shown that exercise has a regulatory role on oxidative stress reaction, there is a lack of study on the relationship between a special form of exercise—CIMT and oxidative stress reaction. Comparing the effect of CIMT and UE on post-stroke limb function and exploring underlying mechanisms of action carry major implications for clinical practice and research. Our study demonstrated that regulating oxidative stress reaction is an important mechanism of CIMT induced neural recovery.

Our study has some limitations. There are great differences in the exercise therapy applied in the RCT and the animal study, regarding the exercise type. In the clinical trial, we only follow up with the patients for 3 weeks, and the long-term effects of CIMT and UE remain to be studied. Moreover, we did not analyze whether CIMT had different effects on different types of cerebral infarction. Previous study indicated that the spontaneous, functional prognosis varies among different types of strokes [43]. The RCT suggested that CIMT influences the emotion of patients with cerebral infarction. However, we did not conduct emotional assessment in animal study.

## 5. Conclusions

In conclusion, CIMT is more effective than UE in improving upper limb motor function and reducing muscle spasm in patients with cerebral infarction than UE, but CIMT may cause the patients to feel depressed. Moreover, both CIMT and UE are beneficial to limb function recovery and limit the infarct expansion in MCAO/R rats, but CIMT was more effective than UE. Oxidative stress reaction has an essential role in regulating the CIMT induced benefits.

## Figures and Tables

**Figure 1 brainsci-13-00004-f001:**
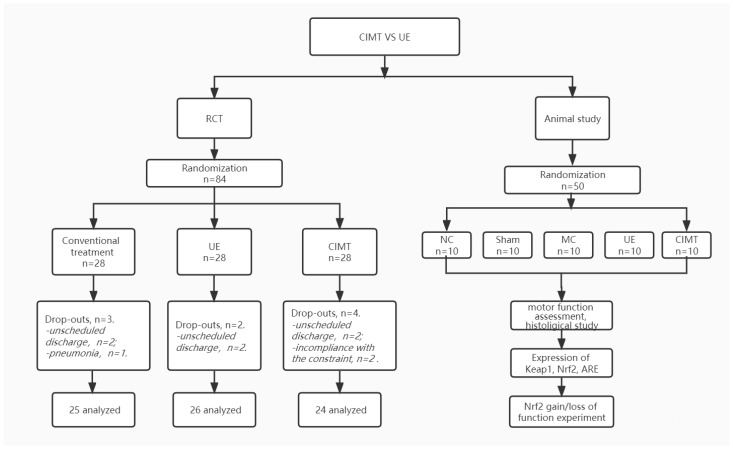
Flowchart of the clinical randomized controlled trial and animal experiment.

**Figure 2 brainsci-13-00004-f002:**
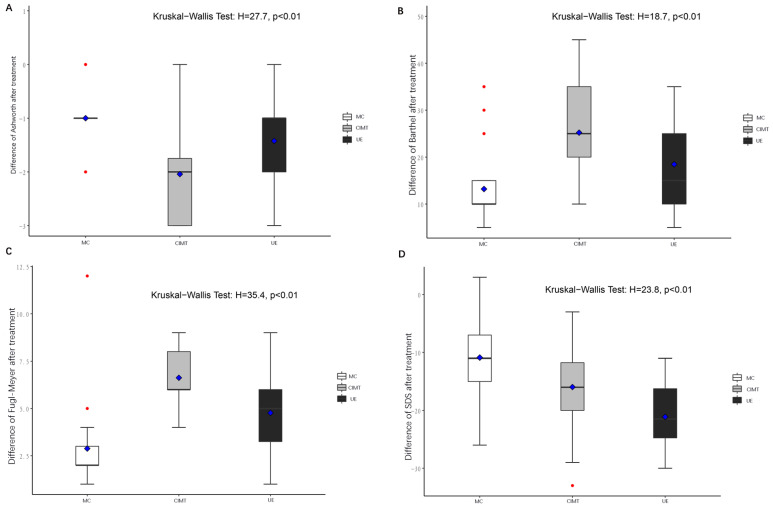
CIMT was more effective than UE in improving limb function in stroke patients. (**A**) change of Ashworth score before and after the treatment in each group; (**B**) change of Barthel index before and after the treatment in each group; (**C**) change of Fugl–Meyer score before and after the treatment in each group; (**D**) change of SDS before and after the treatment in each group. Significance was calculated with the Kruskal–Wallis test. MC: model control; CIMT: constraint-induced movement therapy; UE: unconstraint exercise.

**Figure 3 brainsci-13-00004-f003:**
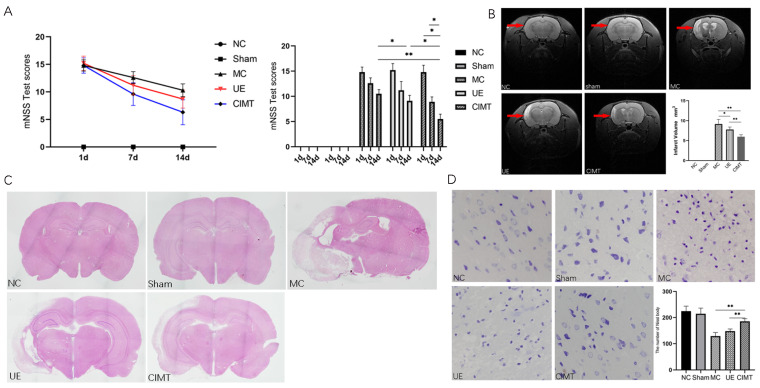
CIMT was more effective than UE in improving limb function in MCAO/R rats. (**A**) the mNSS test of rats in each group on days 1, 7, and 14; (**B**) coronal images of brain MRI performed after the intervention. The infarction regions were indicated by red arrows in the MRI, and the infarction volume of each group was calculated; (**C**) hematoxylin–eosin staining of coronal sections of brains in each group; (**D**) Nissl staining of the brain tissues in each group. The number of Nissl bodies was calculated. NC: normal control; MC: model control; CIMT: constraint-induced movement therapy; UE: unconstraint exercise. *: *p* < 0.05; **: *p* < 0.01.

**Figure 4 brainsci-13-00004-f004:**
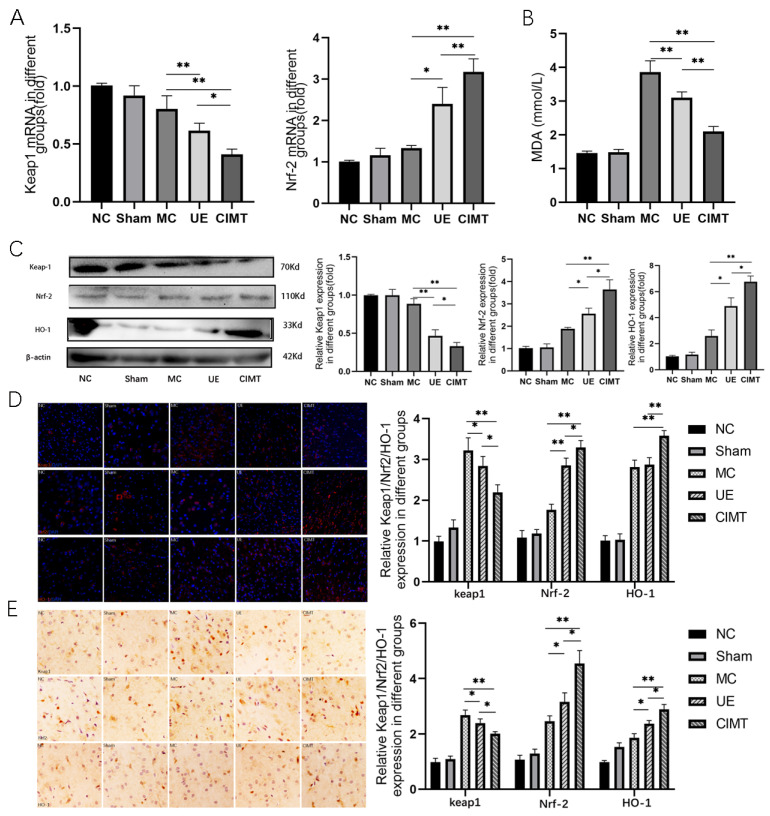
Keap1-Nrf2 signaling pathway is linked to the CIMT induced benefits. (**A**) the relative expression of Keap1 and Nrf-2 mRNA detected by qRT-PCR; (**B**) the concentration of MDA in the peri-infarct zone of the brain; (**C**) the relative expression of Keap1, Nrf-2, and HO-1 detected by Western blot analysis. Representative images are shown; (**D**) the relative expression of Keap1, Nrf-2, and HO-1 detected by immunofluorescence. Representative images are shown; (**E**) the relative expression of Keap1, Nrf-2, and HO-1 detected by immunohistochemistry. Representative images are shown. NC: normal control; MC: model control; CIMT: constraint-induced movement therapy; UE: unconstraint exercise. *: *p* < 0.05; **: *p* < 0.01.

**Figure 5 brainsci-13-00004-f005:**
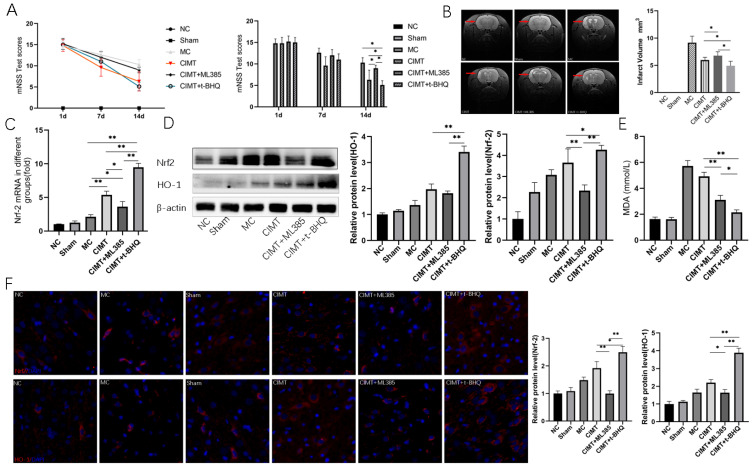
Nrf2 has a pivotal role in regulating CIMT induced benefits. (**A**) the mNSS test of rats in each group on days 1, 7, and 14; (**B**) coronal images of brain MRI performed after the intervention. The infarction regions were indicated by red arrows in the MRI, and the infarction volume of each group was calculated; (**C**) the relative expression of Nrf-2 mRNA detected by qRT-PCR; (**D**) the relative expression of Nrf-2 and HO-1 proteins detected by Western blot analysis. Representative images are shown. (**E**) the concentration of MDA in the peri-infarct zone of the brain; (**F**) the relative expression of Nrf-2 and HO-1 detected by immunofluorescence. Representative images are shown. NC: normal control; MC: model control; CIMT: constraint-induced movement therapy; UE: unconstraint exercise. *: *p* < 0.05; **: *p* < 0.01.

**Table 1 brainsci-13-00004-t001:** Primers for real-time polymerase chain reaction.

Genes	Directions	Sequences
cap3	Forward	CTACCGCACCCGGTTACTAT
	Reverse	TTCCGGTTAACACGAGTGAG
catenin	Forward	GACAAGCCACAGGACTACAAGAA
	Reverse	CGTATCCACCAGAGTGAAAAGAA
wnt3a	Forward	GAATGGTCTCTCGGGAGTTTGC
	Reverse	CAGCAGGTCTTCACTTCGCAAC
Nrf2	Forward	GCCTTCCTCTGCTGCCATTAGTC
	Reverse	TCATTGAACTCCACCGTGCCTTC
keap1	Forward	TGCTCAACCGCTTGCTGTATGC
	Reverse	TCATCCGCCACTCATTCCTCTCC

**Table 2 brainsci-13-00004-t002:** The demographic characteristics.

	N	Sex		Age (Years)	Disease Duration (Months)	Time between Disease Onset and Rehabilitation Treatment (Months)
		F	M			
Control	25	9	16	61.72 ± 9.09	4.04 ± 1.74	2.56 ± 1.56
CIMT	24	9	15	61.38 ± 10.15	3.33 ± 1.71	2.17 ± 1.13
UE	26	8	18	60.12 ± 11.31	3.12 ± 1.66	1.96 ± 1.28
*p*		0.706		0.848	0.141	0.318

CIMT: constraint-induced movement therapy; UE: unconstraint exercise; F: female; M: male.

**Table 3 brainsci-13-00004-t003:** The baseline scores of Ashworth, Barthel, and SDS.

	FMA-UE	Barthel	SDS
Control	57.36 ± 1.32	56.80 ± 10.50	58.24 ± 5.25
CIMT	57.23 ± 1.21	56.73 ± 10.38	59.08 ± 6.95
UE	57.58 ± 1.50	57.08 ± 8.59	57.58 ± 4.61
*p*	0.773	0.936	0.456

CIMT: constraint-induced movement therapy; UE: unconstraint exercise.

**Table 4 brainsci-13-00004-t004:** The scores of Ashworth.

The Baseline Scores				The Score after the Interventions			
Scores	Control (n)	UE (n)	CIMT (n)	Scores	Control (n)	UE (n)	CIMT (n)
0	0	0	0	0	5	4	4
1	0	0	0	1	6	8	16
2	6	5	5	2	15	13	4
3	17	13	12	3			
4	3	7	7	4			
total (n)	26	25	24	total (n)	26	25	24

**Table 5 brainsci-13-00004-t005:** The scores of FMA-UE, Barthel, and SDS before and after the interventions.

	FMA-UE			Barthel			SDS		
	MC	CIMT	UE	MC	CIMT	UE	MC	CIMT	UE
Baseline	57.36 ± 1.32	57.23 ± 1.21	57.58 ± 1.50	56.80 ± 10.50	56.73 ± 10.38	57.08 ± 8.59	58.24 ± 5.25	59.08 ± 6.95	57.58 ± 4.61
Week 3	60.24 ± 230	64.21 ± 0.78	62.00 ± 2.04	70.00 ± 8.660	82.29 ± 7.07	75.19 ± 5.91	47.36 ± 7.04	43.13 ± 5.05	36.46 ± 3.43
Changes	2.88 ± 2.128	6.63 ± 1.527	4.77 ± 2.006	13.20 ± 7.343	25.21 ± 9.264	18.46 ± 8.92	−10.88 ± 6.65	−15.96 ± 7.25	−21.12 ± 5.36
*p*	<0.001	<0.001	<0.001	<0.001	<0.001	<0.001	<0.001	<0.001	<0.001

CIMT: constraint-induced movement therapy; UE: unconstraint exercise.3.2. CIMT was more effective than UE in improving limb function in MCAO/R rats.

## Data Availability

The data are available upon request to the corresponding author.

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
