# Peer review of "Comparison of the Effects of Constraint-Induced Movement Therapy and Unconstraint Exercise on Oxidative Stress and Limb Function—A Study on Human Patients and Rats with Cerebral Infarction"

_brainsci, 2022, doi:10.3390/brainsci13010004_

Round 1

Reviewer 1 Report

Reviewer comments

Main text

Introduction

The first sentence requires a citation

Methods

The inclusion criteria should come in passage form not in bullet points

Please remove the following criteria: Have unilateral limb dysfunction, with a score of brief Fugl-120 Meyer ≥ 56 (85% of total score). It is misleading. The next criteria: extension of wrist joint > 20 °; extension of metacarpophalangeal joint and interphalangeal joint of the thumb, and at least two of  the rest four fingers > 10 ° suffice since they indicate mild to moderate impairment in motor function

The following criteria too are not good criteria for CIMT: shoulder joint passive flex-126 ion and abduction > 90 °, external rotation > 45 °; elbow extension ≤ 30 °; forearm pronation and supination > 45 °; and the static standing posture can be maintained for at least 2 minutes. Can you explain why they are needed?

Participant or legally authorized representative give signed informed consent. Providing consent is not an inclusion criterion.

Randomization

Sign not signed

Page 4, line 152: opened not open

Page 4, Line 155: constraint induced movement therapy not modified induced-constraint movement

Intervention

The conventional therapy was not standardized. You said it was given at the physical therapists’ discretion. Don’t you think that this could be a confounding factor?

Discussion

The first paragraph of the discussion should recap the main findings of the study. Then you go ahead and discuss the findings.

Conclusion

The conclusion needs to be reworked. In the conclusion, you need to make an argument and then warrant it.

General comment

Please proofread your manuscript for language correction

Author Response

Response to Reviewers

Reviewers #1

Introduction

  1. The first sentence requires a citation

Response: Thank you for the suggestion. We have cited the following reference for this statement.

GBD 2019 Stroke Collaborators. Global, regional, and national burden of stroke and its risk factors, 1990-2019: a systematic analysis for the Global Burden of Disease Study 2019. Lancet Neurol. 2021 Oct;20(10):795-820. doi: 10.1016/S1474-4422(21)00252-0. Epub 2021 Sep 3. PMID: 34487721; PMCID: PMC8443449.

Methods

  1. The inclusion criteria should come in passage form not in bullet points

Response: Thank you. We revised it in the text. 

  1. Please remove the following criteria: Have unilateral limb dysfunction, with a score of brief Fugl-120 Meyer ≥ 56 (85% of total score). It is misleading. The next criteria: extension of wrist joint > 20 °; extension of metacarpophalangeal joint and interphalangeal joint of the thumb, and at least two of the rest four fingers > 10 ° suffice since they indicate mild to moderate impairment in motor function

Response: thank you for the comments. We respectfully disagreed with the reviewer. The first criterion- Have unilateral limb dysfunction, with a score of brief Fugl-120 Meyer ≥ 56 -specified the limb dysfunction to be unilateral, not bilateral. And a score of brief Fugl-120 Meyer ≥ 56 quantified mild to moderate motor function impairment. Although the following criteria indicate mild to moderate impairment as well, they were set to ensure the participants were eligible for CIMT intervention (e.g., ensure they can perform the tasks of CIMT: eating, ball throwing, dominoes, playing chess or cards, writing, and sweeping the floor).

For clarity, we modified the description as follows:

Also, included patients should have unilateral but not bilateral limb dysfunction, with a score of Fugl-Meyer ≥ 56 (85% of total score) indicating mild to moderate motor impairment. Furthermore, all potential participants were screened whether the included participants can perform the tasks involved in CIMT using the following criteria: extension of wrist joint > 20 °; extension of metacarpophalangeal joint and interphalangeal joint of the thumb, and at least two of the rest four fingers > 10 °, and the movement can be repeated three times within 1 min; shoulder joint passive flexion and abduction > 90 °, external rotation > 45 °; elbow extension ≤ 30 °; forearm pronation and supination > 45 °; and the static standing posture can be maintained for at least 2 minutes [24].

  1. The following criteria too are not good criteria for CIMT: shoulder joint passive flex-126 ion and abduction > 90 °, external rotation > 45 °; elbow extension ≤ 30 °; forearm pronation and supination > 45 °; and the static standing posture can be maintained for at least 2 minutes. Can you explain why they are needed?

Response: These criteria were set to ensure the participant can perform the CIMT tasks (e.g., eating, ball throwing, dominoes, playing chess or cards, writing, and sweeping the floor),

  1. Participant or legally authorized representative give signed informed consent. Providing consent is not an inclusion criterion.

Response: thank you for pointing it out. We have removed it from the text.

Randomization

  1. Sign not signed

Response: thank you for the reminder. We revised it in the text.

  1. Page 4, line 152: opened not open

Response: thank you for the reminder. We revised it in the text.

  1. Page 4, Line 155: constraint induced movement therapy not modified induced-constraint movement

Response: we are sorry for the typo. We revised it accordingly in the text.

Intervention

  1. The conventional therapy was not standardized. You said it was given at the physical therapists’ discretion. Don’t you think that this could be a confounding factor?

Response: we agree with the reviewer that it may be a confounding factor. However, for ethical consideration, it is inappropriate to change the patients’ routine treatments for the reason of participating the scientific research. And more importantly, those treatments are highly tailored to individuals in routine clinical practice, which can be a common confounding factor difficult to avoid especially for studies in the field of rehabilitation medicine. To minimize its influence, the operators received standardized training.

Discussion

  1. The first paragraph of the discussion should recap the main findings of the study. Then you go ahead and discuss the findings.

Response: in our previous version of manuscript, we presented the main findings of our study in the second last paragraph of the discussion section. In the revised manuscript, we moved this paragraph to the beginning part of discussion.

Conclusion

  1. The conclusion needs to be reworked. In the conclusion, you need to make an argument and then warrant it.

Response: thank you for the suggestion. we modify the conclusion section so it reads: In conclusion, CIMT is more effective than UE in improving upper limb motor function, reducing muscle spasm in patients with cerebral infarction than UE; but CIMT may cause the patients to feel depressed. Moreover, both CIMT and UE are beneficial to limb function recovery and limit the infarct expansion in MCAO/R rats, but CIMT was more effective than UE. Oxidative stress reaction has an essential role in regulating the CIMT induced benefits.

General comment

  1. Please proofread your manuscript for language correction

Response: we used language editing service for the revised manuscript. Thank you for the suggestion.

Reviewer 2 Report

Wang et al., conducted a comprehensive study to investigate the effect of CIMT on motor recovery in both animal and human stroke. The study provides unique insights into the mechanisms of effects associated with CIMT. However, the current version of the manuscript is not well-written. More details about the experiments need to be reported. The conclusion is not properly drawn.

1. In the abstract, a brief definition of unconstraint training is needed. I think most conventional rehab treatments are unconstraint. The authors also need to mention if the UE is dose-matched to CIMT – dose matters in CIMT research. Besides, I don’t know what the primary outcome of the study is. It seems to me the study should focus on the motor domain, but the conclusion highlighted the effect on depression as well. It looks messy.

2 In the introduction, why did the authors decide to measure oxidative stress reaction in this study? What is the linkage between ROS and CIMT?

Treadmill training here is decided for the upper extremity?

3 Ethical approval number needs to be included in the method. Please state if the registration of the clinical trial is done or not.

4 What is brief FMA?

5 Section: Interventions in the RCT. Based on the description, I think the authors may just use high-dose (dose-matched) TOT to represent the second group.

6 The conventional rehab here looks very comprehensive – how did the authors make sure all groups receive comparable treatments? Any pharmacological treatments?

7 In fig 1, the animal study part looked confusing. Why did the author put the hypothesis “CIMT superior to UE” in the middle? The authors should mention how many samples were used in the animal studies in the figure.

8 statistical analyses. The authors should run a 2-way ANOVA instead of doing inter/intra analysis separately.

9 In the results, please consider comment 8 and rewrite the results.

10 In the results, the Ashworth scale is not a numerical scale – don't use mean and SD.

11 In the discussion, the first para should be a summary of the main findings. Please do not say the background knowledge like PNF and Bobath training here again.

12, “in basic research, there is a lack of study on the relationship 453 between CIMT and oxidative stress reaction.” If so, why did the author study ROS in this CIMT study, please see my comment 2 as well.

13 The conclusions look not matched with the conclusion in the abstract.

Author Response

Response to Reviewers

Reviewers #2

Comments and Suggestions for Authors

  1. Wang et al., conducted a comprehensive study to investigate the effect of CIMT on motor recovery in both animal and human stroke. The study provides unique insights into the mechanisms of effects associated with CIMT. However, the current version of the manuscript is not well-written. More details about the experiments need to be reported. The conclusion is not properly drawn.

Response: Thank you for reviewing our manuscript. Please see our response below to your comments. 

  1. In the abstract, a brief definition of unconstraint training is needed. I think most conventional rehab treatments are unconstraint. The authors also need to mention if the UE is dose-matched to CIMT – dose matters in CIMT research. Besides, I don’t know what the primary outcome of the study is. It seems to me the study should focus on the motor domain, but the conclusion highlighted the effect on depression as well. It looks messy.

Response: thank you for your kind comments. In the abstract, we added definitions for CIMT and unconstraint training. It reads: Most conventional post-stroke rehabilitation treatments do not involve imposed constraint of the unaffected limb. In contrast, Constraint-Induced Movement Therapy (CIMT) comprises of massed task practice with the affected limb, and constraint of the unaffected limb. CIMT is a promising rehabilitation technique used for motor recovery of affected limb after stroke, but its effectiveness and mechanism are not fully understood.

We tried to make the dose even in UE and CIMT groups. We took into account both length and intensity of training to make their doses as equal as possible.

The primary outcomes of the study are motor functions. We are also interested in the psychological effects of these training mode, because patient’s emotional state could have some effects on their compliance. Depression is not a primary outcome. We revised this part into: Motor functions of the limb are primary outcomes of the RCT measured using Brief Fugl-Meyer upper extremity score (FMA-UE), Ashworth score and Barthel scale. Psychological influence of CIMT and UE was also examined using Self-Rating Depression Scale (SDS).

  1. In the introduction, why did the authors decide to measure oxidative stress reaction in this study? What is the linkage between ROS and CIMT?

Response: there are many mechanisms regulating recovery after cerebral infarction, such as nerve regeneration, inflammatory recovery and stress injury. Previous studies have shown that exercise therapy is closely related to oxidative stress. Exercise can promote the balance between the rate of ROS production and the rate of antioxidant scavenging ROS. Given the relationship between exercise and ROS, it is intriguing how different exercise modes influence oxidative stress reaction after stroke.

We added some discussion of the link between ROS and exercise in the introduction section. Thank you.

  1. Treadmill training here is decided for the upper extremity?

Response: thank you for your comment. A training equipment special for upper extremity of rat is not available in our lab. So, we chose treadmill training. For rats, walking on treadmill requires synergetic movement of the forelimbs and hindlimbs. In our opinion, treadmill training can simulate a training for the upper limb, although hindlimbs are also involved.

  1. Ethical approval number needs to be included in the method. Please state if the registration of the clinical trial is done or not.

Response: thank you for your comments. Our study was approved by ethical committee of our Institution (Year 2020, No. 1011).

Also, study was registered on Chinese Clinical Registry (ChiCTR2100042339).

We provided this information in the revised manuscript.

  1. What is brief FMA?

Response:thank you for your comments. Here we mean the subscale of FAM- Fugl-Meyer Upper Limb Assessment. For clarity, we changed it in the text (Fugl-Meyer Upper Limb Assessment).

  1. Section: Interventions in the RCT. Based on the description, I think the authors may just use high-dose (dose-matched) TOT to represent the second group.

Response: thank you for the suggestion. We agreed with the reviewer that the intervention described in the second group is essentially the same with task-oriented training. However, we used the name “unconstraint” with the intention to contrast with constraint training.

  1. The conventional rehab here looks very comprehensive – how did the authors make sure all groups receive comparable treatments? Any pharmacological treatments?

Response: we agreed with the reviewer that the treatments patients received was comprehensive and they could be a confounding factor. However, for ethical consideration, it is inappropriate to change the patients’ routine treatments for the reason of participating the scientific research. And more importantly, those treatments are highly tailored to individuals in routine clinical practice, which can be a common confounding factor difficult to avoid especially for studies in the field of rehabilitation medicine. To minimize its influence, the operators received standardized training.

No special pharmacological treatments were given to the participants. Routine medicine includes anticoagulant therapy (oral aspirin, 100mg/d).   

  1. In fig 1, the animal study part looked confusing. Why did the author put the hypothesis “CIMT superior to UE” in the middle? The authors should mention how many samples were used in the animal studies in the figure.

 Response: Thank you. We revised the figure 1.

  1. statistical analyses. The authors should run a 2-way ANOVA instead of doing inter/intra analysis separately.

Response: thank you for your comments. We respectfully disagree with the reviewer. The data is not normally distributed so two-way ANOVA is not appropriate. So nonparametric analysis, Wilcoxon matched pairs signed rank test for intra-group analysis, and KW test for inter-group analysis were used. We tried Scheirer-Ray-Hare (SRH) analysis (nonparametric counterpart of two-way anova), the interaction was statistically significant, so SRH analysis is not appropriate here. Therefore, Wilcoxon matched pairs signed rank test and KW test can be used for intra- and inter- analysis.

11.In the results, please consider comment 8 and rewrite the results.

Response: Thank you. We revised it in the results.

  1. In the results, the Ashworth scale is not a numerical scale – don't use mean and SD.

Response: thank you for pointing it out. Indeed, the data from Ashworth scale are ranked data, and using mean and SD to describe the data is inappropriate. When choosing the statistical analysis method, we used Kruskal Wallis H analysis for Ashworth data. Kruskal Wallis H analysis worked with ordinal categorical data. But in table 4, we presented the data in a wrong way (mean ± SD). In the revised manuscript, we generated another table illustrating the change of distributions of Ashworth grades among three groups before and after the interventions.

  1. In the discussion, the first para should be a summary of the main findings. Please do not say the background knowledge like PNF and Bobath training here again.

Response: in our previous version of manuscript, we presented the main findings of our study in the second last paragraph of the discussion section. In the revised manuscript, we moved this paragraph to the beginning part of discussion.

14.“in basic research, there is a lack of study on the relationship 453 between CIMT and oxidative stress reaction.” If so, why did the author study ROS in this CIMT study, please see my comment 2 as well.

Response: thank you for your comments. The relationship between exercise and oxidative stress reaction has been well studied. CIMT is a special form of task-oriented exercise training, it is worthy of investigating whether CIMT can as well influence ROS, and thus neural and limb functional recovery.

  1. The conclusions look not matched with the conclusion in the abstract.

Response: we modified the conclusion section. Thank you for the comments.

Reviewer 3 Report

The authors  conducted a dual study: a randomized clinical trial in 75 stroke patients and an experimental animal study to evaluate the effects of constraint-induced movement therapy (CIMT) and unconstraint exercise (UE) on limb dysfunction in stroke survivors and to investigate the relationship between CIMT/UE and oxidative stress reaction. The authors found that, compared with UE, CIMT has more significant effects on improving upper limb function and reducing spasms in humans. However, UE was more effective on improving patients’ emotion than CIMT. In addition, CIMT can significantly improve the limb motor function of rats in the MCA/R model than UE,  which involved oxidative stress response, neural regeneration and cell apoptosis. The study is potentially interesting but can be improved if the following considerations are addressed: 

        1. Please describe the acronyms used (ex: RCT, in Introduction) 
        2. Due to single-center and small size of the clinical sample, the authors should clearly mention as another limitation of the present study that the clinical findings in the RCT represent “preliminary findings”.  
        3. It would be interesting to know if the authors found differences in the results obtained in patients with limb weakness secondary to small vessel disease versus weakness secondary to extensive non-lacunar cerebral infarction or cerebral hemorrhage, since lacunar infarcts usually have a better spontaneous functional prognosis than other non-lacunar strokes (see and add this recent reference: Int J Mol Sci 2022; 23, 1497).  
        4. It would be interesting to include in the text a comment that cerebral atrophy is a new emerging feature of cerebrovascular diseases, and this atrophy of the gray matter is usually progressive and documented mainly in patients with acute stroke of the lacunar type (see and comment on the study published in Cerebrovasc Dis 2010; 30: 157-166). The relation between the effects of CIMT or UE and limb function in relation to cerebral atrophy should be further investigated in future research.  

Author Response

Response to Reviewers

Reviewers #3

  1. Please describe the acronyms used (ex: RCT, in Introduction)

Response, thank you for the suggestion. We have checked that full names have been provided prior to the acronyms in the revised manuscript.

  1. Due to single-center and small size of the clinical sample, the authors should clearly mention as another limitation of the present study that the clinical findings in the RCT represent “preliminary findings”.

Response, Thank you. We revised it in the Discussion.

  1. It would be interesting to know if the authors found differences in the results obtained in patients with limb weakness secondary to small vessel disease versus weakness secondary to extensive non-lacunar cerebral infarction or cerebral hemorrhage, since lacunar infarcts usually have a better spontaneous functional prognosis than other non-lacunar strokes (see and add this recent reference: Int J Mol Sci 2022; 23, 1497).

It would be interesting to include in the text a comment that cerebral atrophy is a new emerging feature of cerebrovascular diseases, and this atrophy of the gray matter is usually progressive and documented mainly in patients with acute stroke of the lacunar type (see and comment on the study published in Cerebrovasc Dis 2010; 30: 157-166). The relation between the effects of CIMT or UE and limb function in relation to cerebral atrophy should be further investigated in future research. 

Response: we thank the reviewer for the suggestions. We carefully read the references provided. It is interesting to further investigate how CIMT works with different type of cerebral infarction, as the spontaneous functional prognosis varies among different types of strokes. We added this information in the discussion section in the revised manuscript incited the suggested reference.

Round 2

Reviewer 2 Report

The authors have well addressed my comments. The limitations of the study have been well acknowledged.